# Discovering Glioma Tissue through Its Biomarkers’ Detection in Blood by Raman Spectroscopy and Machine Learning

**DOI:** 10.3390/pharmaceutics15010203

**Published:** 2023-01-06

**Authors:** Denis Vrazhnov, Anna Mankova, Evgeny Stupak, Yury Kistenev, Alexander Shkurinov, Olga Cherkasova

**Affiliations:** 1Laboratory of Laser Molecular Imaging and Machine Learning, Tomsk State University, 634050 Tomsk, Russia; 2V.E. Zuev Institute of Atmospheric Optics SB RAS, 634055 Tomsk, Russia; 3Faculty of Physics, Lomonosov Moscow State University, 119991 Moscow, Russia; 4Novosibirsk Research Institute of Traumatology and Orthopedics n.a. Ya.L. Tsivyan, 630091 Novosibirsk, Russia; 5Institute on Laser and Information Technologies, Branch of the Federal Scientific Research Centre “Crystallography and Photonics” of RAS, 140700 Shatura, Russia; 6Faculty of Automation and Computer Engineering, Novosibirsk State Technical University, 630073 Novosibirsk, Russia

**Keywords:** optical methods of tissue study, glioma tissue, Raman spectroscopy, machine learning, U87 glioblastoma

## Abstract

The most commonly occurring malignant brain tumors are gliomas, and among them is glioblastoma multiforme. The main idea of the paper is to estimate dependency between glioma tissue and blood serum biomarkers using Raman spectroscopy. We used the most common model of human glioma when continuous cell lines, such as U87, derived from primary human tumor cells, are transplanted intracranially into the mouse brain. We studied the separability of the experimental and control groups by machine learning methods and discovered the most informative Raman spectral bands. During the glioblastoma development, an increase in the contribution of lactate, tryptophan, fatty acids, and lipids in dried blood serum Raman spectra were observed. This overlaps with analogous results of glioma tissues from direct Raman spectroscopy studies. A non-linear relationship between specific Raman spectral lines and tumor size was discovered. Therefore, the analysis of blood serum can track the change in the state of brain tissues during the glioma development.

## 1. Introduction

The most commonly occurring malignant brain tumors are gliomas, and among them is glioblastoma multiforme (GBM), which accounts for 14.3% of all tumors and 49.1% of malignant tumors [1,2,3]. GBM is the most aggressive, invasive, and undifferentiated type of tumor, and has been designated grade IV by the World Health Organization [4,5]. GBM is among the deadliest neoplasms.

One reason for the poor outcome of glioblastoma is a late-stage diagnosis, since most of the existing methods of noninvasive diagnostics, such as magnetic resonance imaging (MRI) [6,7,8] and computer tomography [9], are ineffective for diagnosing small-size tumors. Optical methods, such as fluorescence imaging [10], multiphoton microscopy [11], photoacoustic imaging [12], optical absorption spectroscopy [13,14,15], Raman spectroscopy and imaging [16,17], and terahertz (THz) imaging [18,19,20] are widely used for direct detection of glioma tissues spectral fingerprints, including glioma’s molecular biomarkers, under cancer tissue surgery resection. Similar methods are very sensitive to tissue chemical content, but have too low a penetration depth to in vivo noninvasive glioma detection. Tissue biopsies can be taken to study cellular and molecular composition. It is an invasive and traumatic procedure.

The tumor development leads to the release of various substances and tumor cells into circulation [21]. Blood, similar to other body fluids, has been shown to contain circulating tumor cells, extracellular vesicles, circulating tumor nucleic acids, proteins, and metabolites, the analysis of which provides information about the cancer development [22]. The highest concentrations of glioma markers are found in cerebrospinal fluid. The biomarkers have also been found in other biofluids. The discovery of cancer biomarkers in biofluids is called liquid biopsy, which has been intensively developed in recent decades [23,24]. Liquid biopsy provides the opportunity to detect cancer instead of a standard biopsy. Control of glioma molecular markers in body fluids has a potential to become equivalent to studying glioma tissues. It is important that similar molecular markers can appear in biofluids at the early stage, opening a way to an early diagnosis [25,26,27,28]. MicroRNAs and proteins were shown to be the most promising glioma biomarkers [22]. Plasma concentrations of alpha-2-glycoprotein, C-reactive protein, and C9 complement component show a significant positive correlation with tumor size [29]. It was established that the following metabolic pathways are disrupted in glioma: taurine/hypotaurine, D-glutamine/D-glutamate, alanine/aspartate/glutamate, glycine/serine/threonine, and pyruvate metabolism [30]. Blood plasma of patients with primary brain tumors exhibits similar changes in all tumor types: increased glucose and pyruvate, decreased citrate, succinate, and glutamine, increased creatine and decreased creatinine. Phenylalanine and tyrosine were both increased, especially in GBM patients [31]. In serum, cysteine was found at higher levels in GBMs, while lysine and 2-oxoisocaproic acid showed higher levels in oligodendrogliomas [32]. Five metabolites (uracil, arginine, lactate, cystamine, and ornithine) allowed significant distinguishing between high- and low-grade glioma patients [33]. Concentrations of 2-hydroxyglutarate enantiomers, currently a recognized gliomas biomarker, are elevated by orders of magnitude in gliomas harboring mutations in isocitrate dehydrogenase1/2 (IDH1/2) [34]. 

Each type of biological tissue has an individual molecular composition and, thus, a unique spectral profile [14,15,35]. A set of such individual states of functional groups of nucleic acids, proteins, lipids, and carbohydrates makes it possible to characterize the specific tissues, which ultimately makes it possible to isolate disease markers. Vibrational spectroscopy methods include infrared (IR) absorption spectroscopy and Raman spectroscopy. IR spectroscopy and Raman are complementary techniques, which differ in the way the molecular vibrations are excited and detected [15]. Significant progress has been made in the diagnosis of human gliomas by IR spectroscopy of blood serum [14,36,37,38,39,40,41]. However, it was shown that Raman spectroscopy can provide a more precise chemical composition of a biological sample and can be more effective than IR spectroscopy due to the higher sensitivity to C–C, C=C, and C≡C bonds [42,43]. A high number of carbon functional groups was observed in biological tissue and fluids. Raman spectroscopy allows for the discovery of different compounds present in a biological sample by the characteristic frequencies of individual chemical groups of these molecules. It is a powerful analytical technique that can be applied in the diagnosis of cancer, including gliomas [15,17,44,45]. Depending on the stage of tumor development, the greatest changes in tissues’ Raman spectra are observed in the fingerprint spectral region (600–1800 cm^−1^), and in the high-wavenumber spectral region (2400–3800 cm^−1^). The first region is associated with the contribution of the vibration of molecular group of proteins, nucleic acids, amino acids, and lipids, and the second, mainly proteins and lipids, and also allows for determining the water content, which varies significantly in carcinogenic samples. 

Raman spectroscopy of blood serum combined with advanced chemometrics provides the discrimination between samples of breast cancer patients and healthy volunteers [46,47], and between samples from cirrhotic patients with and without hepatocellular carcinoma [48]. A comparison of Raman spectroscopy data of cancer tissues, cancer patient serum, and cancer cell lines was carried out for ovarian, prostate, pancreatic, and breast cancer [44]. The identical spectral features corresponding to specific cancer biomarkers had been found in tissues and serum. Similar results for gliomas have not been presented yet. 

In this work, we are establishing a link between glioma’s tissues’ molecular markers and the similar markers in blood serum, using Raman spectroscopy and machine learning. 

## 2. Materials and Methods

### 2.1. Samples

Mouse blood serum was used for spectroscopic measurements. The study was conducted on 60 male mice of the immunodeficient SCID line at the age 6–7 weeks. The model of orthotopic xenotransplantation of the U87 human glioblastoma cells into mice was used [49,50]. A total of 5 μL of the U87 MG cell suspension (500,000 cells per one injection) was introduced in the subcortical brain structure through a hole in the animal’s cranium. Animals from the control group were injected in a similar manner with 5 µL of the culture medium. The tumor size was measured using a horizontal 11.7 Tesla MRI tomograph (Biospec 117/16; Bruker, Billerica, Waltham, MA, USA) [49]. Each experimental group had a corresponding control group. There were 10 mice in each group. The animals were removed from the experiment by decapitation on days 7, 14, and 21 after injection. Blood samples were collected, serum was separated, and frozen at −80 °C. The study was carried out in accordance with the EU Directive 2010/63/EU and the ARRIVE 2.0 guidelines, and approved by the Inter-Institutional Commission on Biological Ethics at the Institute of Cytology and Genetics, Siberian Branch of the Russian Academy of Sciences (Permission #78, 16 April 2021).

### 2.2. Raman Spectroscopy

DXR Raman microscope (Thermo Fisher Scientific, Waltham, MA USA) with a magnification of 10×, excitation wavelengths of 532 nm, and range 80–4000 cm^–1^ was used. Each sample of blood serum was a droplet with a volume of 10 μL placed on a special aluminum plate [51,52]. This plate had identical holes in the form of a funnel with a diameter of 5 and a depth of 2 mm (see Appendix A). This form of the plate leads to a uniform distribution of the sample upon drying. We can assume that the distribution of samples in each well and the film thickness of the sample after drying will be the same. The measurements were carried out after complete drying of the sample, which took place for 20 min at room temperature. The Raman spectrum of dried blood serum was recorded at three points in the center of the well and averaged (see Appendix A). Each spectrum was averaged over 300 scans to remove cosmic ray noise. 

### 2.3. Machine Learning Pipeline

The implemented machine learning pipeline is shown in Figure 1.

At preprocessing step, Raman spectra were normalized by maximal value, in order to be able to compare spectral intensities. Also, we applied adaptive iteratively reweighed penalized least squares method to remove background noise [53]. A Python library BaselineRemoval contains several methods to remove the noise, caused by fluorescence in Raman spectra, and according to our previous findings [54], the airPLS algorithm [51] has a good performance. Principal component analysis (PCA) was used to extract informative features [55,56]. 

Support vector machine (SVM) [57], random forest (RF) [58], and XGboost [59] were used to build prediction data models. The ten-fold cross-validation was used [60]. The prediction data model performance was estimated using a confusion matrix (see Table 1 for reference), where true positive is the number of correctly diagnosed U87 mice, true negative is the number of control mice labeled as healthy, false negative and positive values are respective errors of classification.

The matrix members can be combined using receiver operating characteristic–area under curve (ROC AUC) based on true positive rate (TPR) versus false positive rate (FPR) values [61]. Here, TPR=TP/(FP+TN), and FPR=FP/(FP+TN). The ROC curve is a plot of TPR (vertical axis) versus FPR (horizontal axis). Whenever the area under the ROC curve is close to 1 or 0, the data model has a perfect quality. If it is close to 0.5, then the performance is equal to random guessing. Combination of k-fold cross-validation and ROC AUC analysis can visualize mean and standard deviation of ROC curve [60]. 

## 3. Results

### 3.1. Characteristics of Experimental Animals

In the dynamics of the U87 glioblastoma development, the tumor size was 2.6 ± 0.4, 10.6 ± 1.8, and 89.6 ± 11.5 mm^3^ (means ± standard error) on days 7, 14, and 21 after tumor cells’ injection, respectively. Figure 2 shows that there is a non-linear increase in tumor size from day 7 to day 21 of the experiment. 

### 3.2. Raman Spectra Analysis

Below, in Figure 3, we present preprocessed Raman spectra of U87 and control groups for all weeks of the experiment.

As depicted in the Figure 3, standard deviation of Raman spectra decreases over time, and this can be related to the tumor growth [50,52]. The major changes in brain tissue metabolism are associated with tumor growth in U87-3 groups, hence, we expect to see regular, rather than random, peaks on the Raman spectra (see Figure 3c). 

The groups’ separability was visually estimated by PCA method. Thorough analysis is required to plot all possible combinations of more than 3000 principal components (PCs) (see Appendix A). Therefore, we focus only on 10 principal components, which cover more that 75% of variance. The visualization of the best group’s separability by PCA is presented in Figure 4. 

As shown in Figure 4, the tumor and control groups differ, yet have intersections. Interestingly, Raman data groups are separated in principal components planes with low explained variance. That points to the minor difference in relative intensity. Also, for the second week of the experiment, there is a good separability, yet some samples are closer to the control group. We believe that the progression of the glioma is lower for them.

Next, we tested separability of the tumor group versus control group for each week by: SVM, RF, and XGBoost with and without application of PCA. The results for the third week of the experiment are presented in Figure 5, Figure 6 and Figure 7. The results for the first and second weeks of the experiment are presented in Appendix A. Table 2 shows the total data for all groups and methods.

We can see from Table 2 that SVM shows the best value for the U87-1 and U87-3 groups with the application of PCA, and for the U87-2 group without the application of PCA. For RF, we can see opposite patterns for the U87-1 and U87-2 groups compared to SVM. Application of PCA does not affect ROC AUC analysis for XGBoost. 

### 3.3. Informative Feature Selection

The results of the informative feature selection for the third week of the experiment are shown in Figure 8. The same results for the first and second weeks can be found in Appendix A.

All methods point to the Raman shifts near 1000 cm^−1^ and another one near 3000 cm^−1^, yet RF and XGBoost marks 600 cm^−1^ and 2600 cm^−1^, respectively. The variation can be explained by a high dimensionality of the data, so it should be verified by the respective mean spectra.

It should be noted that molecule or molecular band identification requires both central line position and band height/width parameters, and a good important feature selection algorithm should choose both. From this perspective, SVM feature ranking is reasonable, because Raman shifts near the central line are also significant. There is a special point about feature selection in random forests and extreme gradient boosting methods [62]. If variables are highly correlated, which is the case for the central line and surrounding band, the algorithm either chooses randomly between them (RF) or focuses on of them (XGBoost). What we can see in Figure 8 is not incorrect results, but specifics of the method, which should be taken into account, which is that is a variation of the central line position is possible. Additionally, we have filtered out features with low information gain for better representation.

We also measured tumor volume growth during the experiment (see Figure 2). As we can see, the change in volume is a non-linear function, so we applied a non-linear regression model based on RF (Python module sklearn.ensemble.RandomForestRegressor with default parameters except regularization parameter ccp_alpha was set to 0.015 to avoid overfitting), with Raman spectra as independent and tumor volume as dependent variable. The quality of RF regressor was estimated by mean and standard deviation of absolute error (MAE) and determination coefficient (R^2^). Validation was performed by repeating ten-fold CV 20 times. This method also allows selecting the most informative features (see Figure 9). To avoid plotting more the 3000 features with importance values, we select only those with an absolute value greater than 0.02. Such a small value of importance is due to the large number of features. The results are the following: MAE = 25.3 ± 7.4, R^2^ = 0.919. Informative Raman shifts are: 417, 420, 818, 1746, 2807, 3039, 3212, and 3563 cm^−1^. Table 3 shows the correspondence between the Raman shifts for the constructed regression model, molecular vibrations, and the corresponding substances [44,63,64,65,66,67,68,69,70,71,72,73,74,75].

## 4. Discussion

The main idea of this paper is to estimate dependency between glioma tissue and biomarkers in blood serum, revealed by Raman spectroscopy. We used the most common model of human glioma, when continuous cell lines such as U87, derived from primary human tumor cells, are transplanted intracranially into the mouse brain [49,76]. This model is characterized by rapid glioma growth following the injection of tumor cells into the subcortical structures of the brain. We have a controlled growth of the tumor and can trace the change in blood composition in the dynamics of tumor development. 

The modeling of glioblastoma with orthotopic transplantation of glioblastoma cells in the brains of laboratory mice is associated with many traits of traumatic brain injury syndrome [77,78]. It is especially clear in the first week of the experiment. The U87 cell suspension was introduced in the subcortical brain structure through a hole in the animal cranium. Animals from the control group were injected in a similar manner with 5 µL of the culture medium. When injected, brain injury occurs and an inflammation reaction develops, both in control and experimental mice. Therefore, in the first week of the experiment, the development of the inflammation reaction prevails over the development of the tumor [79,80]. The intensity of the characteristic peaks in the fingerprint range of 400–1700 cm^−1^ is higher in control animals and it is the same in high-frequency range of 2700–2950 cm^−1^ (see Figure 2a). Opposite trends are seen for the second and third weeks of the experiment (see Figure 2b,c): the intensity of the Raman peaks is higher in U87 mice. 

We study the separability of the experimental and control groups for different glioma stages after the introduction of tumor cells by machine learning methods and try to discover most informative Raman spectral bands for this separation. SVM, RF, and XGboost methods were chosen because they have good generalization ability and additionally allow the selection of informative features. Figure 5, Figure 6, Figure 7 and Appendix A, as well as Table 2, show that SVM has a better generalization ability, so removing noise with PCA has a positive role for SVM. For RF and XGBoost, on the contrary, the fewer the number of features, the worse their learning. For group U87-3, quite good results are obtained with and without PCA, because there are obvious differences and PCA does not have an influence on them. 

We would like to focus on the principal component (PC) selection for visualization. PCA sorts PCs in a descending order, so the first PCs correspond to the largest amount of variance in data. It is not necessarily related to the groups’ separability [81]. In our case, the first several PCs do not affect group separability, so we do not describe them thoroughly.

At the next stage, we identified informative spectral features, according to which the experimental and control groups were divided at each stage of the experiment. The use of three methods makes it possible to establish differences with the highest degree of reliability (see Figure 8, Appendix A). As can be seen from the frequency distribution, Raman spectroscopy makes it possible to follow the change in the contribution of certain metabolites at each stage of the experiment with an increase in tumor size. It is shown earlier that attenuated total reflectance Fourier-transform infrared (ATR-FTIR) spectroscopy coupled with supervised learning methods allows for identifying glioma patients with tumor volumes of 0.2 cm^3^ [40]. In our study, tumor volumes vary from 0.002 to 0.089 cm^3^ at different stages of the experiment, that is, several orders of magnitude lower than in [40].

We applied a non-linear regression model to select the most informative Raman spectra features associated with tumor growth (see Figure 9 and Table 3). As can be seen from Table 3, the frequencies found by us are in good agreement with the Raman spectra of glioma tissues presented by other authors [44,63,64,65,66,67,68,69,70,71,72,73,74,75]. It should be understood that the Raman spectrum of a sample is a superposition of vibrations of all substances present in the sample [15]. One vibration may be due to the contribution of several substances present in the sample. Vice versa, one substance can correspond to a complete set of peaks [82]. The discrepancy in the identification of informative Raman shifts can be explained by two reasons. First, it is computational. Machine learning uses optimization methods, hence, an approximate solution is sought and there can be several solutions. The stability of the approximate solution is improved by averaging. Such averaging can produce bias in informative Raman shifts [83]. Second, it is physical. Different physical conditions [84] and concentrations of substances, such as salt, can lead to changes in the position of Raman shifts [85]. It has also been shown that, depending on the glioma grade, the tissue microenvironment affects the shift of Raman bands by up to 38% [86]. 

According to Figure 9 and Table 3, with the progression of the tumor, an increase in the contribution of lactate, tryptophan, fatty acids, lipids, and water molecules is observed. Using the method of THz spectroscopy, which is known to be sensitive to the state of water in biological samples [87], we have previously shown that in the U87-3 group, the state of water changes, and the contribution of bound water molecules increases [79]. The same patterns are seen in the most informative Raman spectral features for each stage of the experiment. Thus, we show that the analysis of blood serum can track the change in the state of brain tissues during the development of glioblastoma.

The presented results confirm the possibility of glioblastoma detection by Raman spectra analysis of blood serum. This study has several limitations. First of all, the number of samples in each group is not large (10 per group). Though a proper validation of constructed machine learning models is made, a comparison to a reference method is required, namely, MRS, to determine metabolites in tumor and blood tests in order to find the quantity of these metabolites in blood. Second, in our previous work [50], we show that THz spectroscopy can be useful for the same task, so the combination of these modalities by ensemble machine learning techniques can be performed. Third, interesting data separation into groups of the mixed tumor and control samples can be seen on PCA visualization figures. This can be caused either by physiological processes of tumor development or by external factors, such as inflammation, due to the injection of the culture medium. The reason for such behavior will be investigated in further research. 

The strong sides of this study are the following. The conducted experiments with a small animal glioblastoma model were performed in strict accordance with the research protocol and can be reproduced. State-of-the-art machine learning methods were used to verify the separability of the classes for each week of the experiment. The validity of the obtained results is tested by k-fold CV techniques and ROC AUC analysis. Raman spectra informative ranges were selected, which allowed us to speed-up further analysis. We can focus only on narrow bands, which reduces spectra acquisition time. A non-linear relationship between specific Raman spectral lines and tumor size was discovered. This information coupled with THz spectroscopy data allows the possibility of pointing to a metabolic profile associated with glioblastoma development.

## 5. Conclusions

The major goal of the work was discovering glioma through the detection of its biomarkers in the blood by Raman spectroscopy. In the work, the dried blood serum of mice was studied during the U87 glioblastoma development. It is a commonly accepted laboratory model of human glioma implemented under controlled conditions. Machine learning methods were applied to analyze the Raman spectra and identify the most informative frequencies. Thus, we show that the analysis of blood serum can track the change in the state of brain tissues during the glioma development. Raman spectroscopy combined with machine learning allows us to carry out label-free and real-time analysis of brain tissue and blood serum, as well as the differentiation of glioma grades and rapid monitoring of treatment. 

## Figures and Tables

**Figure 1 pharmaceutics-15-00203-f001:**
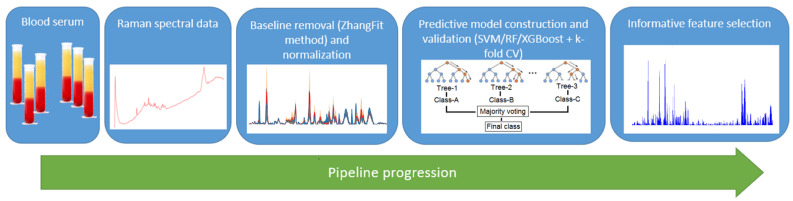
The implemented machine learning pipeline for Raman spectra analysis.

**Figure 2 pharmaceutics-15-00203-f002:**
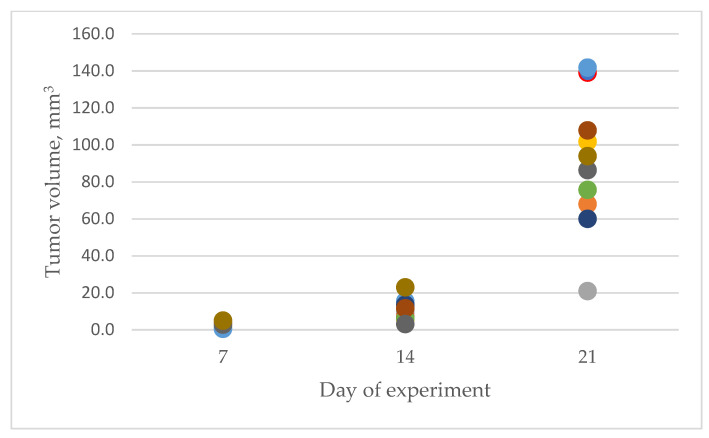
Tumor volume, measured on the 7th, 14th, and 21st day of the experiment.

**Figure 3 pharmaceutics-15-00203-f003:**
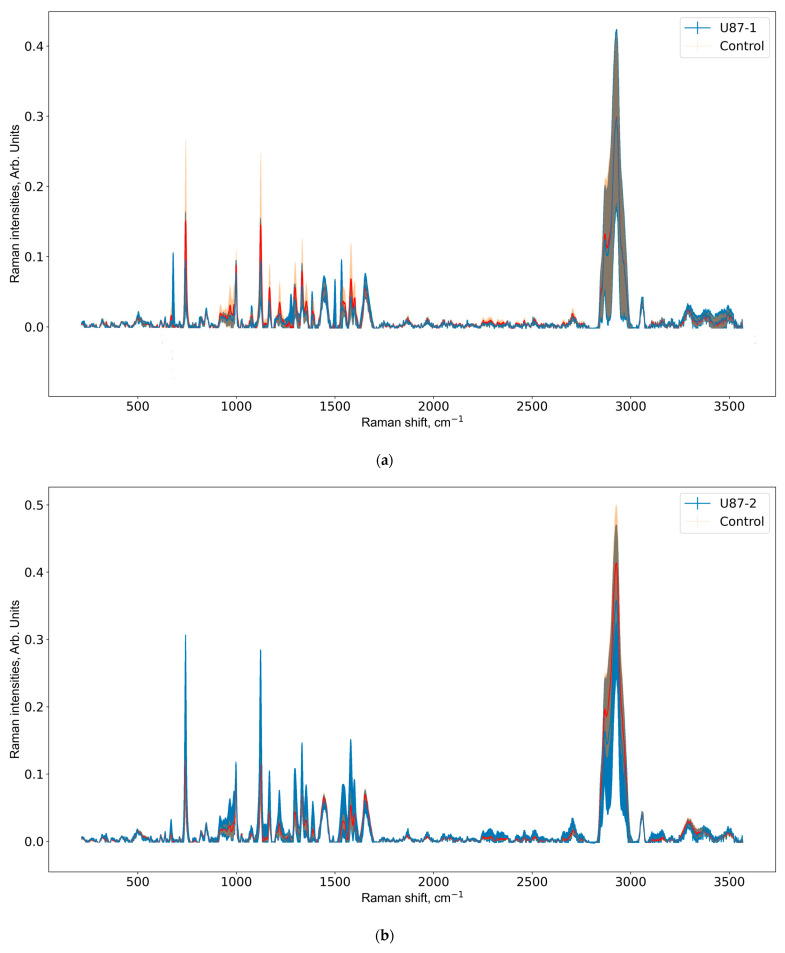
Mean and standard deviation of Raman spectra of U87-1, U87-2, U87-3, and control groups for first (**a**), second (**b**), and third (**c**) weeks of the experiment. Subtracting standard deviation from the mean may lead to negative values, which were then removed.

**Figure 4 pharmaceutics-15-00203-f004:**
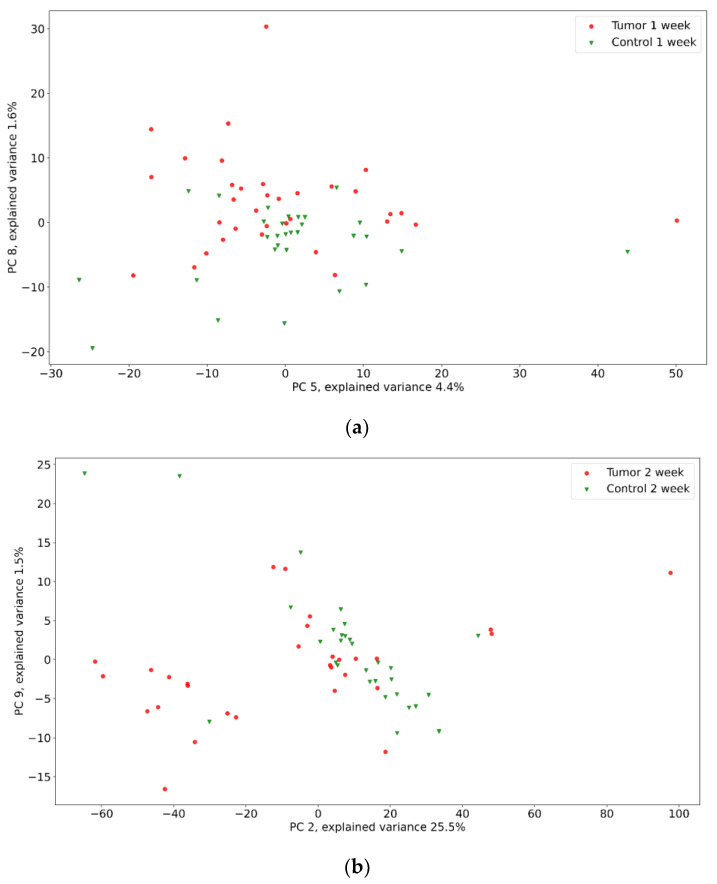
PCA method applied to Raman spectra of tumor and control groups for first (**a**), second (**b**), and third (**c**) weeks of the experiment.

**Figure 5 pharmaceutics-15-00203-f005:**
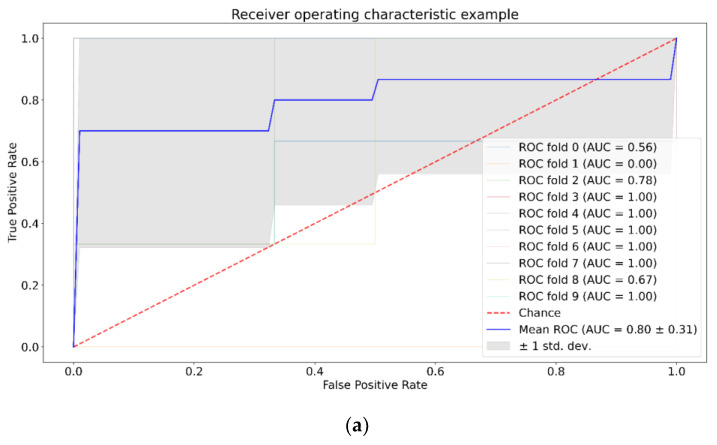
ROC AUC analysis for SVM, third week, no PCA (**a**) and with PCA (**b**).

**Figure 6 pharmaceutics-15-00203-f006:**
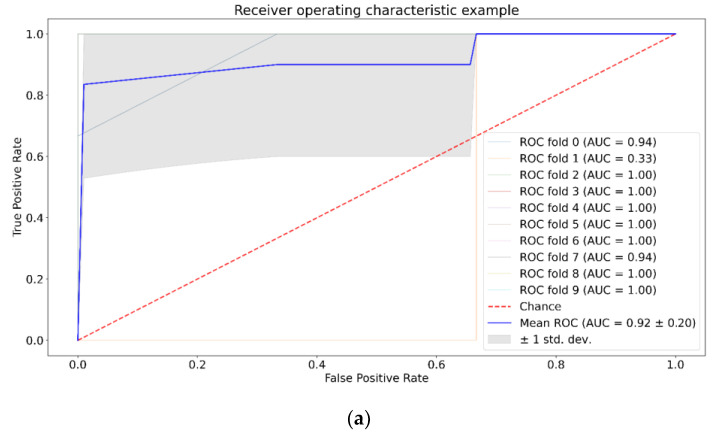
ROC AUC analysis for RF, third week, no PCA (**a**) and with PCA (**b**).

**Figure 7 pharmaceutics-15-00203-f007:**
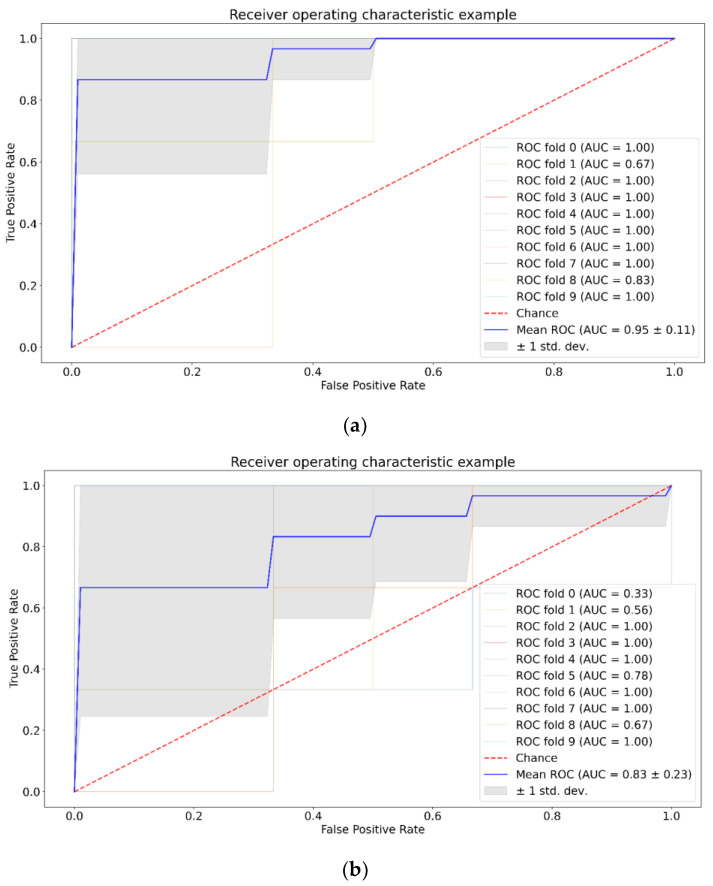
ROC AUC analysis for XGBoost, third week, no PCA (**a**) and with PCA (**b**).

**Figure 8 pharmaceutics-15-00203-f008:**
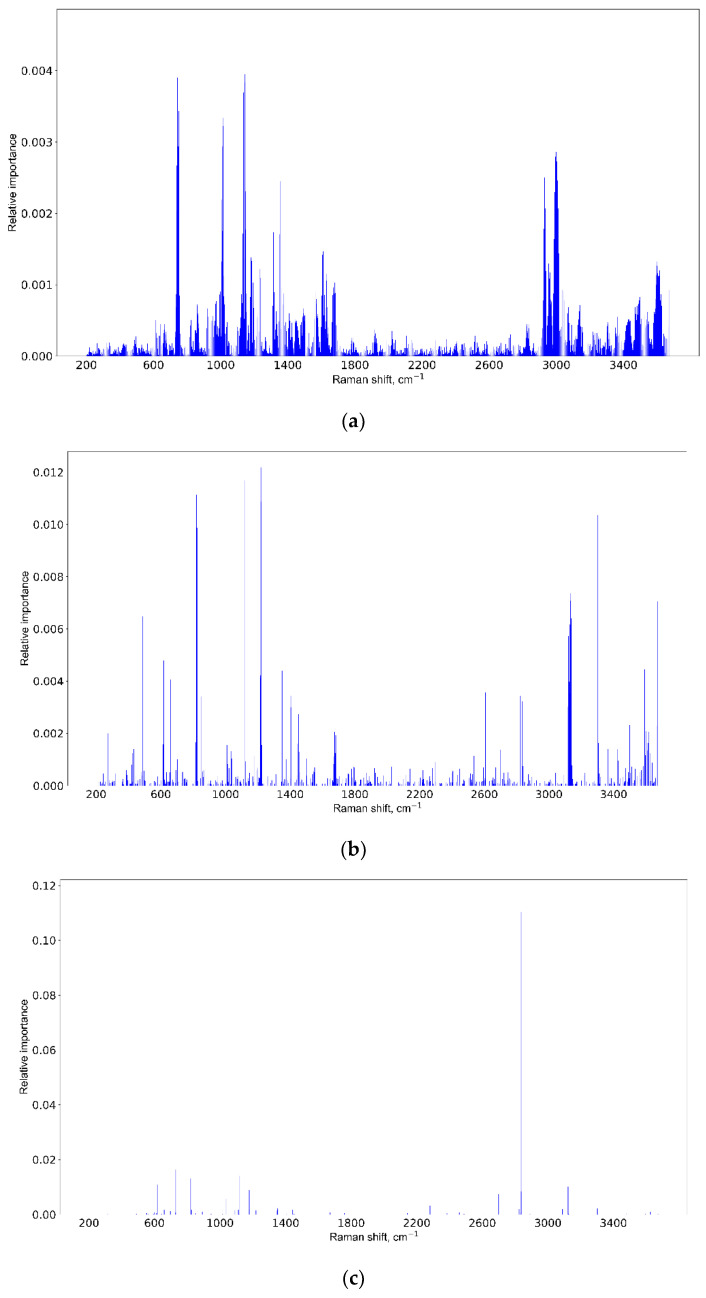
Informative feature selection for the third week of the experiment for SVM (**a**), RF (**b**), and XGBoost (**c**) methods.

**Figure 9 pharmaceutics-15-00203-f009:**
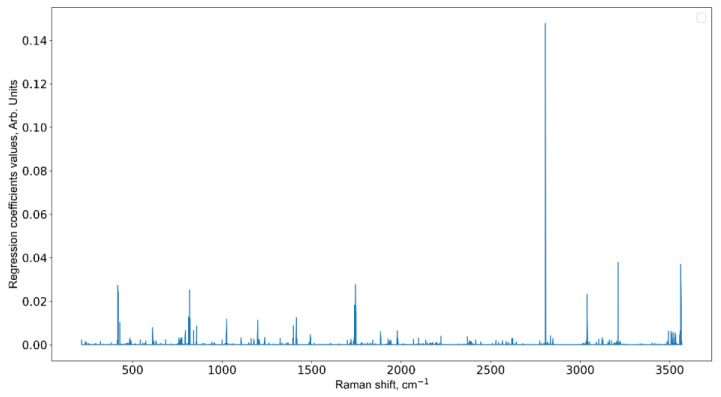
Important features (Raman shifts) for the constructed regression model.

**Table 1 pharmaceutics-15-00203-t001:** Confusion matrix.

	Positive (Model Prediction)	Negative (Model Prediction)
Positive (expert label)	True positives (TP)	False negative (FN)
Negative (expert label)	False positive (FP)	True negative (TN)

**Table 2 pharmaceutics-15-00203-t002:** Mean and standard deviation of AUC for SVM, RF, XGBoost.

	SVM no PCA	SVM with PCA	RF no PCA	PF with PCA	XGBoost no PCA	XGBoost with PCA
U87-1	0.72 ± 0.38	0.91 ± 0.17	0.81 ± 0.29	0.78 ± 0.21	0.92 ± 0.13	0.87 ± 0.19
U87-2	0.80 ± 0.28	0.68 ± 0.32	0.73 ± 0.22	0.78 ± 0.29	0.82 ± 0.0.18	0.72 ± 0.27
U87-3	0.80 ± 0.31	0.85 ± 0.25	0.92 ± 0.20	0.92 ± 0.16	0.95 ± 0.11	0.83 ± 023

**Table 3 pharmaceutics-15-00203-t003:** Raman spectral peak position and assignments of molecular vibrations.

Raman Peaks (cm^−1^)		Assignments		
This Paper Blood Serum	Glioma Brain tissues	Modes	Molecules	References
417 420	391		Protein, lipids Cholesterol/cholesterol ester	[63]
	419			[64]
	431			[65]
818	754	CC, CN2, CH2 rock	Nucleic acids, DNA, Trp,	[66] [67] [65] [65] [68]
			Nucleoproteins, Cyt	
	800	C-C stretching	L-Tryptophane	
	817		Collagen	
	826		Tyr, proline	
	825–827	O-P-O stretch	DNA	
1746	1736	C=O	Lipids	[66]
	1737		Cholesterol	[69]
	1739		Cholesterol ester	[69]
	1740		Protein	[63]
2807	2850	CH2	Lipids, fatty acids	[44,66,70]
3039	3040	-OH CH3 CH3	Water	[71]
	2930		Proteins and DNA	[44]
	2985		Lactate	[72]
	2932		Lipoprotein	[66]
	2939		Lactate, lipoprotein	[66]
3039	3064	NH3	Nucleic acids/ protein, Trp	[66]
3212	3174	NH, CH3	Amid B, L-glutamine	[66]
	3164	str (NH), str (OH)	Amide, water	[73]
	>3200	str (OH)	Water	[74]
3563	3350–3550	OH	Water	[72,75]

## Data Availability

The data presented in this study are available on request from the corresponding author.

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
