# Peer review of "Discovering Glioma Tissue through Its Biomarkers’ Detection in Blood by Raman Spectroscopy and Machine Learning"

_pharmaceutics, 2023, doi:10.3390/pharmaceutics15010203_

Round 1

Reviewer 1 Report

I regret to point out that I do not feel qualified to judge about the technical aspects of this paper with respect to the spectral analyses. I am quite familiar with the characterization based on vibrational spectroscopy, but the spectral calculations reported here are beyond my range and are necessary to understand in order to judge about the validity of the findings. Therefore, I am requesting that another reviewer gets included. A few things that I can point out is that this is a very relevant topic and perhaps a more detailed of studies that previously tried to detect glioblastoma based on blood markers and their challenges can be included in the introduction. Also, a better description of what constitutes controls is needed. I could not find information where control blood specimens were derived from. Also, authors may want to justify their choice of particular controls. Are they healthy blood samples or could they also involve blood samples from animals suffering from another form of cancer? There could be arguments in favor of one or the other. In any case, because I cannot judge about the validity of technical calculations, I am relegating this review to another specialist. 

Author Response

Dear Reviewer,

We would like to thank you for your work with our submission and for your thoughtful and relevant remarks. We have revised our manuscript according to your comments. Please, see the details of this revision below. For your convenience, the main changes are highlighted with the green-colored text in the revised manuscript.

 Response to the Reviewer #1:

 Comment #1: A few things that I can point out is that this is a very relevant topic and perhaps a more detailed of studies that previously tried to detect glioblastoma based on blood markers and their challenges can be included in the introduction.

Authors' Response:  The glioma's molecular marker discovery in body fluids is the most important from the point of view of noninvasive diagnosis and treatment control [1, 2]. Studies of glioma molecular markers have been intensively developed during the past decade using genomics, proteomics, and metabolomics. The additional material about discovered glioma biomarkers in body fluids has been added in Introduction section:

“MicroRNAs and proteins were shown to be the most promising glioma biomarkers [22]. Plasma concentrations of alpha-2-glycoprotein, C-reactive protein, and C9 complement component showed a significant positive correlation with tumor size [29]. It was established that the following metabolic pathways are disrupted in glioma: taurine/hypotaurine, D-glutamine/D-glutamate, alanine/aspartate/glutamate, glycine/serine/threonine, and pyruvate metabolism [30]. Blood plasma of patients with primary brain tumors exhibited similar changes in all tumor types: increased glucose and pyruvate, decreased citrate, succinate, and glutamine, increased creatine and decreased creatinine. Phenylalanine and tyrosine were both increased essentially in GBM patients [31]. In serum, cysteine was found at higher levels in GBMs, while lysine and 2-oxoisocaproic acid showed higher levels in oligodendrogliomas [32]. Five metabolites (uracil, arginine, lactate, cystamine, and ornithine) significantly allowed distinguishing high- and low-grade glioma patients [33]. Concentrations of 2-Hydroxyglutarate enantiomers, currently recognized gliomas biomarker, are elevated by orders of magnitude in gliomas harboring mutations in isocitrate dehydrogenase1/2 (IDH1/2) [34]. “

References:

  1. Figueroa, J. M. Detection of glioblastoma in biofluids. J Neurosurg 2018, 129, 334–340. doi: 10.3171/2017.3.JNS162280.
  2. Touat, M.; Duran-Peña, A.; Alentorn, A.; Lacroix, L.; Massard, C.; Idbaih, A. Emerging circulating biomarkers in glioblastoma: promises and challenges. Expert Rev Mol Diagn 2015, 15(10), 1311-1323, doi: 10.1586/14737159.2015.1087315.
  3. Ali, H.; Harting, R.; de Vries, R.; Ali, M.; Wurdinger, T.; Best, M.G. Blood-Based Biomarkers for Glioma in the Context of Gliomagenesis: A Systematic Review. Front. Oncol. 2021, 11, 665235. doi: 10.3389/fonc.2021.665235.
  4. Miyauchi, E.; Furuta, T.; Ohtsuki, S.; Tachikawa, M.; Uchida, Y.; Sabit, H.; et al. Identification of blood biomarkers in glioblastoma by SWATH mass spectrometry and quantitative targeted absolute proteomics. PLoS ONE 2018, 13(3), e0193799. doi: 10.1371/journal.pone.0193799.
  5. Lee JE, Jeun SS, Kim SH, Yoo CY, Baek H-M, Yang SH (2019) Metabolic profil-ing of human gliomas assessed with NMR. J Clin Neurosci 68:275–280.
  6. Baranovičová, E.; Galanda, T.; Galanda, M.; et al. Metabolomic profiling of blood plasma in patients with primary brain tumours: Basal plasma metabolites correlated with tumour grade and plasma biomarker analysis predicts feasibility of the successful statistical discrimination from healthy subjects - a preliminary study. IUBMB Life 2019, 71(12), 1994-2002. doi:10.1002/iub.2149.
  7. Mörén, L.; Bergenheim, A.T.; Ghasimi, S.; Brännström, T.; Johansson, M.; Antti, H. Metabolomic screening of tumour tissue and serum in glioma patients reveals diagnostic and prognostic information. Metabolites 2015, 5, 502–520.
  8. Zhao, H.; Heimberger, A. B.; Lu, Z.; Wu, X.; Hodges, T. R.; Song, R.; Shen, J. Metabolomics profiling in plasma samples from glioma patients correlates with tumor phenotypes. Oncotarget 2016, 7(15), 20486-20495. doi: 10.18632/oncotarget.7974.
  9. Strain, S.K.; Groves, M.D.; Olino, K.L.; Emmett, M.R. Measurement of 2-hydroxyglutarate enantiomers in serum by chiral gas chromatography-tandem mass spectrometry and its application as a biomarker for IDH mutant gliomas. Clinical Mass Spectrometry 2020, 15, 16–24. doi:10.1016/j.clinms.2019.11.002.

Comment #2: Also, a better description of what constitutes controls is needed. I could not find information where control blood specimens were derived from. Also, authors may want to justify their choice of particular controls. Are they healthy blood samples or could they also involve blood samples from animals suffering from another form of cancer?

Authors' Response: Thank you for your comment.  The following text has been added in manuscript.

The study was conducted on 60 male mice of the SCID (SHO-PrkdcscidHrhr) line at the age 6 - 7 weeks. 5 μL of the U87 MG cell suspension (500000 cells per one injection) were introduced in the subcortical brain structure through a hole in the animal’s cranium. Animals from the control group were injected in a similar manner with 5 µL of the culture medium. Each experimental group had a corresponding control group. There were 10 mice in each group. Intravital 1H magnetic resonance spectroscopy was performed on the anesthetized animals before surgery and on days 7, 14 and 21 after the injection. The experimental and corresponding control groups were examined on the same day. This made it possible to follow the changes in brain structures and metabolites after injection of either tumor cells in experimental mice or culture medium in control animals. The data has not been published yet. The tumor size was measured using a horizontal 11.7 Tesla MRI tomograph (Biospec 117/16; Bruker, Billerica, MA, USA) [35].

Reviewer 2 Report

The manuscript describes the use of Raman microspectroscopy to analyse dried human blood serum and identify potential indicators of brain cancer. The field of biomedical spectroscopy has developed considerably over 20 years, and over the past decade in particular the use of bodily fluids for early stage disease diagnostics has been particularly active. Very prominent in this context has been the work of the group in Scotland in the development of ATR-FTIR spectroscopy of dried serum for the early stage diagnosis of glioma, and subsequent efforts to translate the technique towards clinical applications (https://www.dxcover.com/) .  The work of the group is cited as [14], but it should be discussed more extensively, particularly in the context of how the Raman analysis of the current study aims to, and/or succeeds in improving on this already quite mature approach.

[34] does not specifically deal with "cirrhotic patients with and without hepatocellular carcinoma", but more generally reviews "Quantitative analysis of human blood serum using vibrational spectroscopy", and, in doing so points out some important considerations in the measurement of serum , compared to the measurement of dried serum.

The current  study presents a study of dried blood  or liquid biopsy or serum. In all places, including the title, this should be made clear. A potential significant issue with the measurement of dried blood/serum is the chemical/physical inhomogeneity of the deposit   (Vroman or coffee-ring effect). Whereas ATR-FTIR can integrate over the deposit, it is not clear that the Raman measurement does. Section 2.2. Raman spectroscopy should therefore indicate what the size of the deposit was, and what the spot size of the laser was, at the sample. The three measurements of a given sample should be shown, before and after preprocessing, to give an indication of the intrinsic variability of the measurement.

The spectra of figure 1 (a in particular) have a lot of negative going features, which indicate over correction by the preprocessing procedure. This procedure should be reconsidered.

Relying on PCs as high as 8 or 9 to be able to differentiate between the different serum datasets implies that there is a lot of variability in the lower order PCs much of which could come from the measurement protocol, and the authors should evaluate the PC1, PC2 comparison, to try to identify and eliminate the source of that variability.

In Raman spectroscopy, a feature is a Lorentzian/Gaussian (or mixed) band with a lineshape/linewidth which depends on the chemical composition, the sample environment, and the measurement instrumentation. An appropriate feature selection algorithm should identify such bands, rather than individual wavenumbers. If the response at a wavenumber at  the centre of a band is significant, the ones around it should also.

Raman band assignments should be referenced.

Overall, the potential clinical relevance of the work should be discussed, in the context of other work in the field.

Round 2

Reviewer 1 Report

The authors have revised the publication based on the limited comments I have had, but I am not sure what happened with my request to invite a specialist with knowledge on mathematical processing of the vibrational spectra.

Reviewer 2 Report

The authors have satisfactorily addressed the issuers raised during the initial review.